# The Impact of Corticosteroids on Mortality in Acute Exacerbations of Idiopathic Pulmonary Fibrosis: A Meta-Analysis

**DOI:** 10.3390/arm93020006

**Published:** 2025-03-28

**Authors:** Pier-Valerio Mari, Angelo Coppola, Lorenzo Carriera, Francesco Macagno

**Affiliations:** 1Internal Medicine, San Carlo di Nancy Hospital, 00165 Rome, Italy; 2Pulmonology, San Filippo Neri Hospital, 00135 Rome, Italy; coppolangelo@gmail.com; 3Università Cattolica del Sacro Cuore, Campus of Rome, 00168 Rome, Italy; lorenzocarriera@gmail.com; 4Pulmonology Division, Fondazione Policlinico Universitario A. Gemelli, Istituti di Ricovero e Cura a Carattere Scientifico—IRCCS, 00168 Rome, Italy; macagnofrancesco@gmail.com

**Keywords:** steroids, corticosteroids, acute exacerbation, idiopathic pulmonary fibrosis

## Abstract

**Highlights:**

**What are the main findings?**

**What is the implication of the main finding?**

**Abstract:**

**Background**: Acute exacerbation (AE) of idiopathic pulmonary fibrosis (IPF) is one of the most challenging events in the disease course due to the high mortality despite treatment. The role of corticosteroid treatment in AE-IPF has never been defined, even though it is used in current clinical practice. We performed a meta-analysis to determine the effects of steroid treatment on the acute exacerbation outcomes in idiopathic pulmonary fibrosis (IPF). **Objectives**: To evaluate the impact of steroids on mortality in patients affected by an acute exacerbation of IPF. **Methods**: This meta-analysis was performed in accordance with the PRISMA statement. A systemic literature search was conducted through Google Scholar, Scopus, WoS, PubMed, and JSTOR. Manuscripts from January 2014 to September 2024 were included in the analysis. Articles were included on whether participants had an acute exacerbation of IPF. Regarding the intervention performed, we evaluated the studies in which patients underwent treatment with corticosteroids. As outcomes, studies were included if they analyzed the overall mortality. **Results:** A total of 2156 records were initially identified. Nineteen studies (3277 patients) were ultimately included in the final analysis, comparing 1552 patients who received steroids to 1725 patients without steroids. Steroid treatment poses a higher risk, as suggested by the summary measures (RR of 1.78; CI 1.29–2.76, *p* = 0.00001). **Conclusions**: This meta-analysis investigated the debated role of corticosteroid treatment during acute exacerbation of idiopathic pulmonary fibrosis. Overall, steroid therapy is associated with increased risk. Clinicians should carefully weigh the risks and benefits of corticosteroid therapy in acute exacerbation of IPF.

## 1. Introduction

Idiopathic pulmonary fibrosis (IPF) is a rare interstitial lung disease (ILD) characterized by clinical, radiological, and lung function decline associated with poor survival [1]. Acute exacerbations (AEs) are uncommon events in idiopathic pulmonary fibrosis, even though they may affect between 2% and 16% of patients and are associated with high in-hospital mortality or 30-day mortality. According to the 2016 International Working Group Report [2], AE-IPF is defined by acute respiratory deterioration (less than 1 month) in patients with a previous diagnosis of IPF, along with compatible radiographic abnormalities that other causes cannot explain. Furthermore, exacerbations of fibrosing interstitial disease are also observed in different forms of progressive interstitial lung fibrosis and are consistently associated with a poor prognosis like IPF [3]. To date, the treatment of such exacerbations has been based on high-dose/low-dose/pulse steroid therapy, despite the absence of solid research foundations and based on anecdotal reports, case series, or previous retrospective studies [4]. It also does not seem to have been too long since the PANTHER study [5] demonstrated how the use of steroids and immunosuppressive therapy can disappoint us and is associated with higher mortality, even though it was the standard of treatment, along with the NAC, until 2014 for the management of IPF. Therefore, the role of corticosteroid treatment in AE-IPF has never been entirely defined and many doubts have been raised without clear answers, especially given the difficulty of achieving an adequate sample size for a prospective multi-center study. Although the use of corticosteroids in AE-IPF is not clearly defined, their use remains widespread in practice. According to the *American Journal of Respiratory and Critical Care Medicine* guidelines [6], corticosteroid therapy in AE-IPF is usually employed despite a lack of compelling evidence [4].

To provide more solid evidence and elements, we performed a systematic review and a meta-analysis on the use of steroids in acute exacerbation of IPF compared to those who did not undergo such therapy, with a particular focus on mortality.

## 2. Methods

### 2.1. Search Strategy

This meta-analysis was performed in accordance with the Preferred Reporting Items for Systematic Reviews and Meta-Analyses (PRISMA) statement. A systemic literature search was conducted across five databases (Google Scholar, Scopus, WoS, PubMed, and JSTOR). The search strategy employed was [(“Idiopathic Pulmonary Fibrosis” OR “IPF” OR “Pulmonary Fibrosis” OR “Fibrotic Interstitial Lung Disease” OR “Interstitial Lung Disease” OR “ILD”) AND (“Acute Exacerbation” OR “Acute Flare-up” OR “Acute Episode” OR “Acute Worsening”) AND (“Steroid Treatment” OR “Corticosteroids” OR “Prednisolone” OR “Methylprednisolone” OR “Glucocorticoids” OR “Steroid Therapy”) AND (“Mortality” OR “In-hospital Mortality” OR “Death” OR “Survival Rate” OR “Outcome” OR “Survival Analysis”)] in the article title, abstract, and keywords (descriptors), with equivalent search fields in each database. This study was registered with the International Prospective Register of Systematic Reviews (PROSPERO) (https://www.crd.york.ac.uk/PROSPERO accessed on 15 December 2024) and the registration number is CRD42024621574).

### 2.2. Study Selection Criteria

Articles were included on whether participants had an acute exacerbation of idiopathic pulmonary fibrosis, which is defined as an acute, clinically significant respiratory deterioration of unidentifiable cause [2]. Regarding the intervention performed, we evaluated studies in which patients underwent treatment with corticosteroids. As outcomes, studies were included if they analyzed the overall mortality. The search was limited to the last 10 years, with a time frame from 2014 to 2024. We excluded the following material: reviews, editor letters, opinions, guidelines, regulatory documents, bibliographic selections, preprints, inaccessible full texts, and congress or symposium abstracts. We also excluded duplicate records, irrelevant studies that did not focus on idiopathic pulmonary fibrosis (IPF) or acute exacerbations treated with steroids, and studies outside the time frame of 2014 to September 2024. The PICO framework for this meta-analysis was as follows in Table 1.

### 2.3. Data Extraction and Quality Assessment

Data were independently extracted by two authors and reviewed by a third author. Details regarding the study characteristics and population demographics, along with the intervention and outcomes (RR of mortality and their 95% CI), were collected. The risk of bias for the included studies was assessed using the ROBINS-I tool for non-randomized studies. This tool allows for a detailed evaluation of biases, particularly in retrospective studies, which constitute the majority of studies in this meta-analysis. Due to the observational nature of the included studies, the matching between steroid-treated and control patients was not uniformly reported. In studies where matching was performed, patients were matched based on baseline characteristics such as the disease severity and comorbidities. These factors, which are known to influence the AE-IPF outcomes, were considered in the analysis to minimize the confounding effects.

### 2.4. Statistical Analysis

The statistical analyses were performed using the Cochrane Review Manager Version 5.3. The pooled RR and 95% CI for the overall mortality were calculated with a random effect model, since the data were selected from studies performed by researchers operating independently and the heterogeneity (assessed using the I^2^ value determined by the Q test) was expected to be high. *p* values < 0.05 were considered indicative of statistical significance. We also generated forest plots using Python v. 3.13.1 with the matplotlib library.

## 3. Results

The initial search strategy retrieved 2156 records, with 773 duplicates and 611 irrelevant records that were removed. After screening, 61 full-text articles were assessed for eligibility, and 19 studies were ultimately included in the final PRISMA analysis (Figure 1). Table 2 summarizes the study characteristics and population demographics, along with the interventions and outcomes. The quality assessment of the included studies highlighted variability in the levels of bias across different domains. The selection bias was mostly low, but the performance and detection biases were moderate to high in several studies, which could influence the study outcomes. While several studies included in our meta-analysis did not report detailed information about the clinical severity, comorbidities, and patient demographics (such as age and gender), future studies should incorporate these variables to assess their potential influence on the relationship between corticosteroid use and mortality. A separate analysis was conducted for prospective randomized controlled trials (RCTs), which represent a higher level of evidence compared to retrospective studies. In these studies, corticosteroid use was not associated with a statistically significant reduction in mortality (RR = 1.33, 95% CI 0.87–2.03) (Appendix A). The overall risk of bias indicated that while some studies were at a higher risk due to confounding factors, the majority were rated as having a moderate or low risk of bias (Figure 2), suggesting that the evidence from these studies is reasonably robust. However, caution is advised when interpreting the results, given the potential influence of biases in individual studies.

The funnel plot for the RR demonstrated some degree of asymmetry, suggesting potential heterogeneity among the studies or publication bias (Figure 3). A large amount of heterogeneity was observed between the studies (I^2^ = 89%), supporting the use of a random effect model analysis. The spread of the studies was more evident in the lower precision regions (higher standard error), indicating variability in the effect sizes across studies. The observed asymmetry might be attributed to selective reporting or differences in study designs, populations, or methodologies. However, the overall patterns suggest that the cumulative evidence points toward consistent effect estimates despite the potential bias.

Moreover, the cumulative risk ratio (RR) meta-analysis provides valuable insight into the pooled effect estimates across the included studies. As more studies were added, the cumulative RR progressively increased, indicating variability in the effect size estimates among the studies. The cumulative RR documented a trend with larger fluctuations gradually diminishing as more studies were incorporated, indicating a stabilizing effect estimate. This suggests that the RR analysis reached a point of consistency with sufficient study data, providing a reliable measure of the overall effect (Figure 4).

The impact of corticosteroid treatment in patients affected by acute exacerbation of IPF on mortality is exhibited in Figure 5, where 1552 patients who received steroids are compared to 1725 control patients without intervention. The meta-analysis results revealed a risk ratio (RR) of 1.78 (95% CI: 1.29–2.76) when mortality was considered in the intervention group, and the difference was statistically significant (Z = 16.57 and *p* < 0.0001). Risk ratios (RRs) were calculated for each study to compare mortality between patients treated with corticosteroids and those who were not. Despite the retrospective design of many of the included studies, we used a random effects model to account for the expected variability in the effect sizes across studies. The use of the RR is justified given that most studies reported binary outcomes related to mortality, allowing for this calculation.

## 4. Discussion

The main finding of our meta-analysis is the impact that steroids have on patients with exacerbation of idiopathic pulmonary fibrosis, resulting in an increasing overall mortality. Specifically, the relative risk (RR) of 1.78 means that the group exposed to corticosteroid treatment has a 78% higher risk of death than the unexposed group, with a consistent confidence interval.

This finding was suggested by retrospective studies with good sample sizes and clear study designs regarding the dosage of steroid therapy and whether the treatment was pulse. However, the clinical evidence on this matter was minimal, and stronger scientific evidence was needed, which led to the present meta-analysis.

The strengths of our analysis lie in the advantage offered by the meta-analysis, which allowed us to examine an adequate sample size of patients with exacerbation of idiopathic pulmonary fibrosis (1552 patients who received treatment compared to 1725 in the control group). Moreover, we considered overall mortality as the primary outcome, an unbiased hard endpoint. This outcome is characterized not only by robust statistical significance but also by clear clinical relevance, given the extremely high mortality rate of patients with AE-IPF.

The limitations of our analysis are mainly due to the high heterogeneity, with an I^2^ of 89%, and the variability in the criteria used by the studies to define mortality, ranging from in-hospital mortality to 3-month mortality or 12-month mortality. Furthermore, we express caution regarding the correlation between mortality and steroid dosage. The exact corticosteroid dose (mg) varied significantly, with some treatments involving high doses such as 0.5 g or 1 g of methylprednisolone, while others were more conservative, using doses like 0.1 g per kg. The same limitation can be shared for a pulse short-course treatment compared to a longer treatment during the hospital stay due to the heterogeneity of the studies included. Additionally, it was not possible to obtain a precise picture of the degree of inflammation using biomarkers such as C-reactive protein or D-dimer, and the same applies to the lack of data regarding sputum samples, blood cultures, or bronchoalveolar lavages to rule out or confirm infections. One important limitation of this meta-analysis is the inability to establish clear causality between steroid treatment and mortality, as many included studies did not match the steroid-treated patients with controls based on disease severity. This raises the possibility that more severely ill patients may have been more likely to receive steroids, thus confounding the observed relationship between corticosteroid use and mortality. In future studies, prospective randomized controlled trials (RCTs) with better matching strategies could help clarify the role of steroids in AE-IPF outcomes.

Considering these findings, it is essential to emphasize that although our meta-analysis suggests an association between corticosteroid therapy and increased mortality in AE-IPF patients, the current body of evidence does not allow for definitive clinical recommendations. The predominance of retrospective and heterogeneous studies and the variability in treatment protocols and patient populations limit the certainty with which conclusions can be drawn. Moreover, the possibility of confounding by indication—whereby more critically ill patients may have been preferentially treated with corticosteroids—cannot be excluded.

Therefore, while our results highlight a potential signal of harm associated with corticosteroid use in AE-IPF, they should be interpreted with caution. Clinicians should carefully weigh the risks and benefits of corticosteroid therapy on a case-by-case basis, considering individual patient factors and comorbidities. Importantly, the present meta-analysis underscores the urgent need for well-designed, prospective randomized controlled trials to more conclusively determine the role of corticosteroids in this clinical setting.

## Figures and Tables

**Figure 1 arm-93-00006-f001:**
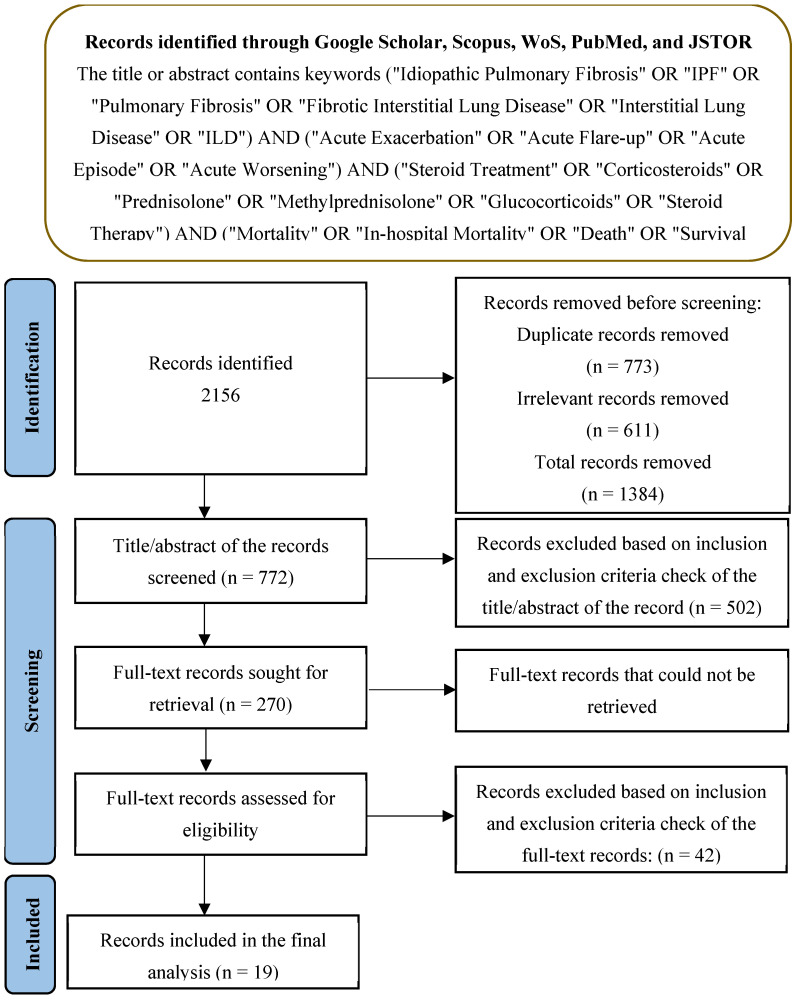
PRISMA framework for the analysis.

**Figure 2 arm-93-00006-f002:**
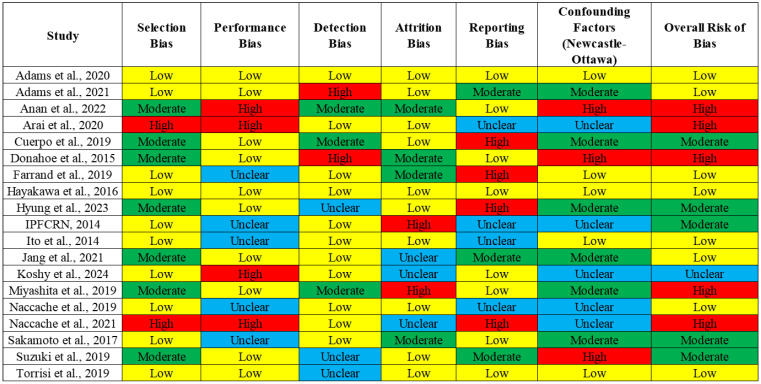
Quality assessment and bias analysis.

**Figure 3 arm-93-00006-f003:**
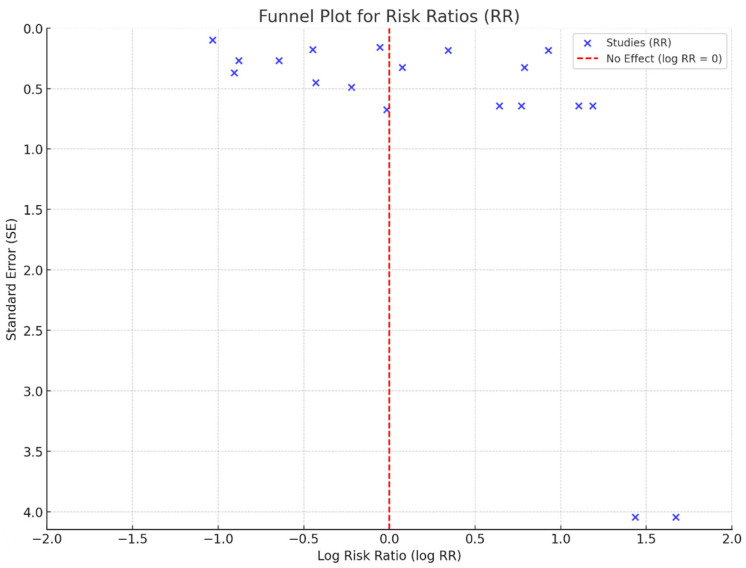
Funnel plot analysis.

**Figure 4 arm-93-00006-f004:**
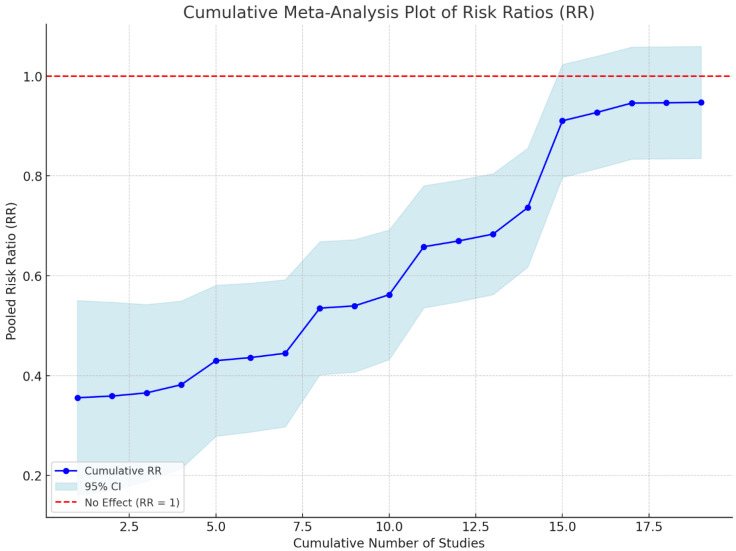
Heterogeneity of the effect sizes across studies. This figure presents the I^2^ statistic to assess the degree of heterogeneity in the studies included in the meta-analysis.

**Figure 5 arm-93-00006-f005:**
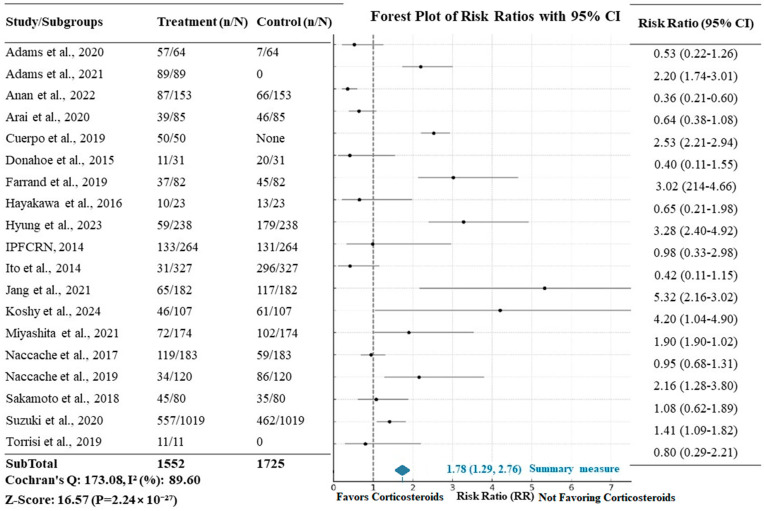
Forest plot showing the relative risk (RR) of mortality with corticosteroid treatment in AE-IPF. The *X*-axis represents the risk ratio (RR), with values > 1 indicating a higher risk of mortality in the corticosteroid group. “n/N” refers to the number of events in each group (treated vs. control) divided by the total number of participants in each group.

**Table 1 arm-93-00006-t001:** PICO framework.

PICO Element	Description
Population	Patients diagnosed with acute exacerbations of idiopathic pulmonary fibrosis (AE-IPF), characterized by a rapid and significant deterioration in respiratory status, typically within one month, who require urgent treatment.
Intervention	Corticosteroid treatment for AE-IPF, including high-dose corticosteroids (such as 1 g methylprednisolone), low-dose regimens, or pulse steroid therapy, administered either intravenously or orally, as part of the clinical management of acute exacerbations.
Comparison	Patients diagnosed with AE-IPF who were not treated with corticosteroids, including those receiving supportive care or other pharmacologic interventions (e.g., immunosuppressive therapies, antifibrotics) for the management of acute exacerbations.
Outcome	The primary outcome was overall mortality, defined as the rate of death occurring during hospitalization (in-hospital mortality) or within a specified follow-up period (long-term mortality), as reported in the included studies.

**Table 2 arm-93-00006-t002:** Studies included in the final meta-analysis.

Author/Year	Title	Study Design	Sample Size
Adams et al., 2020 [7]	Clinical characteristics, outcomes, and associations with mortality in patients admitted with an interstitial lung disease exacerbation	Retrospective analysis	64 cases of AE-ILD identified: 40 IPF and 24 with non-IPF diagnoses.
Adams et al., 2021 [8]	Feasibility and outcomes of a standardized management protocol for acute exacerbation of interstitial lung disease	Retrospective analysis	89 patients
Anan et al., 2022 [9]	Early corticosteroid dose tapering in patients with acute exacerbation of idiopathic pulmonary fibrosis	Retrospective analysis	Multi-center cohort 153 patients; administrative cohort 229 patients.
Arai et al., 2020 [10]	Efficacy of recombinant thrombomodulin for poor prognostic cases of acute exacerbation in idiopathic interstitial pneumonia: secondary analysis of the SETUP trial	Retrospective analysis	85 patients
Cuerpo et al., 2019 [11]	Acute exacerbations of idiopathic pulmonary fibrosis: Does clinical stratification or steroid treatment matter?	Retrospective analysis	50 patients
Donahoe et al., 2015 [12]	Autoantibody-targeted treatments for acute exacerbations of idiopathic pulmonary fibrosis	Prospective trial	11 trial subjects and 20 historical controls.
Farrand et al., 2019 [13]	Corticosteroid use is not associated with improved outcomes in acute exacerbation of IPF	Retrospective analysis	82 patients
Hayakawa et al., 2016 [14]	Efficacy of recombinant human soluble thrombomodulin for the treatment of acute exacerbation of idiopathic pulmonary fibrosis: a single arm, non-randomized prospective clinical trial	Prospective trial	10 patients compared with 13 historically untreated patients.
Hyung et al., 2023 [15]	Pulse versus non-pulse corticosteroid therapy in patients with acute exacerbation of idiopathic pulmonary fibrosis	Retrospective analysis	238 patients
IPFCRN, 2014 [16]	Randomized trial of acetylcysteine in idiopathic pulmonary fibrosis	Prospective trial	264 patients
Ito et al., 2014 [17]	Prophylaxis for acute exacerbation of interstitial pneumonia after lung resection	Prospective trial	31 patients
Jang et al., 2021 [18]	Corticosteroid responsiveness in patients with acute exacerbation of interstitial lung disease admitted to the emergency department	Retrospective analysis	182 patients
Koshy et al., 2024 [19]	Steroid therapy in acute exacerbation of fibrotic interstitial lung disease	Retrospective analysis	107 patients
Miyashita et al., 2021 [20]	Prognosis after acute exacerbation in patients with interstitial lung disease other than idiopathic pulmonary fibrosis	Retrospective analysis	174 patients in total: 102 with AE of IPF and 72 with non IPF
Naccache et al., 2017 [21]	Cyclophosphamide added to glucocorticoids in acute exacerbation of idiopathic pulmonary fibrosis (EXAFIP): a randomised, double-blind, placebo-controlled, phase 3 trial	Prospective trial	183 patients assessed for eligibility, 120 randomly assigned
Naccache et al., 2019 [22]	Study protocol: exploring the efficacy of cyclophosphamide added to corticosteroids for treating acute exacerbation of idiopathic pulmonary fibrosis; A randomized double-blind, placebo-controlled, multi-center phase III trial (EXAFIP)	Prospective trial	120 patients
Sakamoto et al., 2018 [23]	Recombinant human soluble thrombomodulin for acute exacerbation of idiopathic pulmonary fibrosis: a historically controlled study	Retrospective analysis	80 patients in total
Suzuki et al., 2020 [24]	Acute exacerbations of fibrotic interstitial lung diseases	Retrospective analysis	1019 patients in total: 462 with IPF and 557 with other non-IPF ILD
Torrisi et al., 2019 [25]	Possible value of antifibrotic drugs in patients with progressive fibrosing non-IPF interstitial lung diseases	Retrospective analysis	11 patients

## Data Availability

The original contributions presented in this study are included in the article/Appendix A. Further inquiries can be directed to the corresponding author.

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
