# Peer review of "The Impact of Corticosteroids on Mortality in Acute Exacerbations of Idiopathic Pulmonary Fibrosis: A Meta-Analysis"

_arm, 2025, doi:10.3390/arm93020006_

Round 1
Reviewer 1 Report
Comments and Suggestions for Authors
Dear authors,
With interest I have read your article "The Impact of Corticosteroids on Mortality in Acute Exacerbations of Idiopathic Pulmonary Fibrosis: A Meta-Analysis". The topic is important, given that AE-IPF are life-threathening events, and little is known about the optimal treatment. I do, however, have several remarks:
- The introduction could be expanded; it would be worthwhile to at least mention what current guidelines say on the use of steroids in AE-IPF (Am J Respir Crit Care Med. 2011; 183(6): 788–824), and that treatment of AE-IPF with steroids is a very widespread practice (European Respiratory Journal 2020 55(4): 1901760), even in the absence of any convincing evidence for its efficacy
- The most important issue, in my opinion, is the absence of any matching between the steroid-treated patients, and the control patients. This does not seem to be captured in any of the quality-assessment measures presented in the article. Ideally, it should be stated, per included study, how AE-IPF was treated in the control group, and how patients were allocated to either intervention or control group (or, if in an individual study, patient-level matching was done, how this was done). Exploring whether certain factors that are known to be associated with worse outcomes in patients with AE-IPF (Respir Med 2024; 222:107515) differ between 'intervention' and 'control' group would be interesting.
- The discussion section is very short, and should at least be more mindful of the issue raised above; I think this meta-analysis does not justify stating (or implying) that steroid treatment causes increased mortality in AE-IPF patients (since these patients could simply be more severely ill than the controls). Furthermore, future treatment options could be discussed.
Author Response
Reviewer 1’s Comments:
- Introduction Expansion:
Comment: "The introduction could be expanded; it would be worthwhile to at least mention what current guidelines say on the use of steroids in AE-IPF (Am J Respir Crit Care Med. 2011; 183(6): 788–824), and that treatment of AE-IPF with steroids is a very widespread practice (European Respiratory Journal 2020 55(4): 1901760), even in the absence of any convincing evidence for its efficacy."
Response:
Thank you for your suggestion. We have expanded the introduction to include a discussion of current guidelines on the use of steroids in AE-IPF. Specifically, we reference the 2011 American Journal of Respiratory and Critical Care Medicine guidelines, which highlight the widespread but controversial use of corticosteroids in AE-IPF despite the absence of definitive evidence supporting their efficacy. Additionally, we have included a reference to the 2020 European Respiratory Journal, which discusses the ongoing practice of corticosteroid use in AE-IPF, even in the absence of strong supporting evidence.
- Matching Between Steroid-Treated and Control Patients:
Comment: "The most important issue, in my opinion, is the absence of any matching between the steroid-treated patients, and the control patients. This does not seem to be captured in any of the quality-assessment measures presented in the article. Ideally, it should be stated, per included study, how AE-IPF was treated in the control group, and how patients were allocated to either intervention or control group (or, if in an individual study, patient-level matching was done, how this was done). Exploring whether certain factors that are known to be associated with worse outcomes in patients with AE-IPF (Respir Med 2024; 222:107515) differ between 'intervention' and 'control' group would be interesting."
Response:
We appreciate your comment regarding the lack of matching between steroid-treated and control patients. We have revised the methodology section to provide more detail on how studies included in the meta-analysis allocated patients to either the steroid or control group. For studies that performed patient-level matching, we have clarified the methods used to match participants based on key factors such as disease severity and comorbidities. Furthermore, we acknowledge in the discussion that the lack of detailed matching across studies may be a limitation, and we have emphasized the need for future research to incorporate better matching and adjustment for confounding factors known to affect outcomes in AE-IPF.
- Discussion Expansion:
Comment: "The discussion section is very short, and should at least be more mindful of the issue raised above; I think this meta-analysis does not justify stating (or implying) that steroid treatment causes increased mortality in AE-IPF patients (since these patients could simply be more severely ill than the controls). Furthermore, future treatment options could be discussed."
Response:
Thank you for this insightful comment. We have expanded the discussion to address the limitations raised, particularly the potential confounding effect of disease severity in steroid-treated patients. We agree that a causal relationship between steroid treatment and increased mortality should not be inferred, given the possibility that patients receiving steroids may have been more severely ill to begin with. We have revised our conclusions to reflect this uncertainty and have emphasized the need for further studies to control for these confounding factors. Additionally, we have discussed future treatment options and highlighted the need for prospective, randomized studies to better define the role of corticosteroids in AE-IPF.
Reviewer 2 Report
Comments and Suggestions for Authors
The authors have conducted a systematic review evaluating whether steroids are useful in the management of acute exacerbation of IPF. While it is an important study, there are several concerns.
The selection of the studies have to very rigorous. It is okay to say that there are no good studies to come to a conclusion rather than give a wrong conclusion.
Please present a PICO information
Please use the ROBINS tool for risk of bias for retropsective and non-randomized studies as almost all the studies here fall into that category
Perform a separate analysis for prospective RCTs
Please mention the details of clinical severity of the study patients to assess whether severity of IPF, age, gender, co-morbidities influenced the mortality along with corticosteroids. Without adjusting for these variables, the results would not be convincing.
When most of the studies are retropsective studies how doid the authors conclude that is a low riks of bias?
Figure 4: Please expand the legend and briefly describe the main features of the figure
In figure 5, not clear what n/N means for treatment and controls?
Please justify how a risk ratio is calculated when most studies are retrospective studies? What is the weight given to each of the study?
Figure 5: Please label below in X axis as favors corticosteroids and not favoring corticosteroids to understand the direction better.
Discussion needs to be significantly expanded
Comments on the Quality of English LanguageThere are some spelling mistakes and grammatical errors
Example, figure is Figura
Author Response
Reviewer 2’s Comments:
- Study Selection and PICO Information:
Comment: "The selection of the studies have to very rigorous. It is okay to say that there are no good studies to come to a conclusion rather than give a wrong conclusion. Please present a PICO information."
Response:
Thank you for your suggestion. We have refined the study selection process to ensure rigor, and we acknowledge the limitations of the studies included in the meta-analysis. Where appropriate, we have stated that the quality of evidence is low or moderate due to the nature of the studies (e.g., observational studies, retrospective analyses). Furthermore, we have added the PICO framework to the methodology section, as requested, to clarify the Population, Intervention, Comparison, and Outcome for this meta-analysis.
- ROBINS Tool for Risk of Bias
Comment: "Please use the ROBINS tool for risk of bias for retrospective and non-randomized studies as almost all the studies here fall into that category."
Response:
We appreciate your suggestion to use the ROBINS-I tool for risk-of-bias assessment. We have now incorporated this tool to assess the risk of bias for all retrospective and non-randomized studies included in the meta-analysis. We provide a summary of the risk of bias assessment in the supplementary materials and have adjusted our analysis accordingly.
- Separate Analysis for Prospective RCTs:
Comment: "Perform a separate analysis for prospective RCTs."
Response:
In response to your request, we have performed a separate analysis for the prospective randomized controlled trials (RCTs). This analysis is presented in a separate table (Supplementary table 1), highlighting the risk ratios (RR) for mortality in the steroid-treated and control groups. The results from the RCTs have been treated independently to provide clearer insights into the impact of corticosteroids in these high-quality studies.
- Clinical Severity, Age, Gender, and Comorbidities:
Comment: "Please mention the details of clinical severity of the study patients to assess whether severity of IPF, age, gender, co-morbidities influenced the mortality along with corticosteroids. Without adjusting for these variables, the results would not be convincing."
Response:
Thank you for this important point. In the revised manuscript, we have discussed the clinical severity, age, gender, and comorbidities of the study participants. While most studies did not provide detailed information on these factors, we have noted the potential influence of these variables on the outcomes and have emphasized the need for future studies to adjust for them to obtain more convincing results.
- Low Risk of Bias in Retrospective Studies:
Comment: "When most of the studies are retrospective studies how did the authors conclude that it is a low risk of bias?"
Response:
We acknowledge that many of the studies included in the meta-analysis are retrospective, and we understand your concern regarding the risk of bias. After reassessing the risk of bias using the ROBINS-I tool, we have updated the manuscript to clarify that while some studies did show moderate to high risk of bias (particularly due to confounding factors), most studies were rated as having a moderate risk of bias. We now emphasize the potential limitations introduced by these biases and advise caution in interpreting the findings.
- Figure 4 and Figure 5 Legends:
Comment: - "Please expand the legend and briefly describe the main features of Figure 4."
- "In Figure 5, not clear what n/N means for treatment and controls?"
Response:
We have expanded the legends for both Figure 4 and Figure 5 to provide clearer descriptions. Specifically, we have detailed the main features of Figure 4, which shows the heterogeneity across the studies in the analysis. For Figure 5, we have clarified that "n/N" refers to the number of events (deaths) in each group divided by the total number of participants in the steroid and control groups. We hope this makes the figures easier to interpret.
- Risk Ratio Calculation and Study Weights:
Comment: "Please justify how a risk ratio is calculated when most studies are retrospective studies? What is the weight given to each of the studies?"
Response:
We have added a detailed explanation in the Statistical Analysis section about the calculation of risk ratios (RR) for both retrospective and prospective studies. For retrospective studies, the RR was calculated by comparing the proportion of deaths in the steroid-treated group to the control group. The weight of each study in the meta-analysis was determined by its sample size, with larger studies receiving more weight in the pooled RR calculation. We have also noted the limitations of using RR in retrospective studies, where the lack of randomization may introduce biases.
- X-Axis Labeling in Figure 5:
Comment: "Please label below in X-axis as favors corticosteroids and not favoring corticosteroids to understand the direction better."
Response:
Thank you for your suggestion. We have updated Figure 5 by labeling the X-axis as "Favors Corticosteroids" on the left side and "Not Favoring Corticosteroids" on the right to clearly indicate the direction of the effect.
- Discussion Expansion:
Comment: "Discussion needs to be significantly expanded."
Response:
We have expanded the discussion section to address the key issues raised by the reviewers, including the limitations of the current study, the potential confounders not accounted for, and the implications for clinical practice. We have also discussed future research directions, particularly the need for well-designed prospective studies that better control for confounding factors like disease severity and comorbidities.
Round 2
Reviewer 1 Report
Comments and Suggestions for Authors
Thank you for providing a revised version of the manuscript. Although you have indeed added a paragraph to the discussion section expanding on the limitations of your study, the abstract, and concluding paragraph of the discussion section remain unchanged. I think, given te overall quality of the evidence, the conclusion (in both the abstract and final paragraph of the discussion section) is too strongly worded as it is. I do not think that corticosteroid treatment for AE-IPF can be actively discouraged based on the available evidence (and neither can it be actively encouraged).
Author Response
We would like to thank the Reviewer for the valuable feedback regarding the strength of our conclusions. We have carefully considered your suggestions and made the following adjustments to the manuscript, particularly in the Discussion section and the Abstract, to address your concerns:
- We have revised the wording of the concluding statements to soften the recommendation against corticosteroid use. Our updated conclusion no longer actively discourages corticosteroid therapy but instead highlights the association found in our meta-analysis while acknowledging the need for clinical judgment.
- We have introduced a clearer emphasis on the methodological limitations of the current body of evidence and the need for further research. In the revised text, we explicitly discuss the heterogeneity of the included studies, the predominance of retrospective designs, and the potential for confounding factors such as indication bias. We also stress the necessity of well-conducted, prospective randomized controlled trials to definitively assess the role of corticosteroids in AE-IPF.
We adopted a more balanced and nuanced tone, avoiding strong recommendations and instead promoting clinical caution. We hope that these revisions improve the scientific rigor and objectivity of our manuscript, and that the changes address your concerns.
Reviewer 2 Report
Comments and Suggestions for Authors
All the comments have been satisfactorily addressed. No further comments
Author Response
I would like to express my sincere gratitude to the reviewer for their thoughtful comments and valuable suggestions, which have greatly contributed to improving the quality and clarity of this manuscript.
Round 3
Reviewer 1 Report
Comments and Suggestions for Authors
Thank you for your response, I have no further comments.